# Computer Simulation of Composite Materials Behavior under Pressing

**DOI:** 10.3390/polym14235288

**Published:** 2022-12-03

**Authors:** Khrystyna Berladir, Dmytro Zhyhylii, Jiří Brejcha, Oleksandr Pozovnyi, Jan Krmela, Vladimíra Krmelová, Artem Artyukhov

**Affiliations:** 1Department of Applied Materials Science and Technology of Constructional Materials, Sumy State University, 2, Rymskogo-Korsakova St., 40007 Sumy, Ukraine; 2Department of Computational Mechanics named after Volodymyr Martsynkovskyy, Sumy State University, 2, Rymskogo-Korsakova St., 40007 Sumy, Ukraine; 3Faculty of Mechanical Engineering, J. E. Purkyně University in Ustí nad Labem, Pasteurova 1, 400 96 Ustí nad Labem, Czech Republic; 4Faculty of Industrial Technologies in Púchov, Alexander Dubček University of Trenčín, I. Krasku 491/30, 02001 Púchov, Slovakia; 5Academic and Research Institute of Business, Economics and Management, Sumy State University, 2, Rymskogo-Korsakova St., 40007 Sumy, Ukraine

**Keywords:** polymer matrix, simulation, pressing, sintering, process innovation, stress-train curves, ultimate stress, Poisson’s ratio, strain capability, bearing capacity, ANSYS Workbench

## Abstract

Composite materials have a wide range of functional properties, which is ensured by using various technological methods of obtaining both the matrix or fillers and the composition as a whole. A special place belongs to the composition formation technology, which ensures the necessary structure and properties of the composite. In this work, a computer simulation was carried out to identify the main dependencies of the behavior of composite materials in the process of the main technological operations of their production: pressing and subsequent sintering. A polymer matrix randomly reinforced with two types of fillers: spherical and short cylindrical inclusions, was used to construct the finite element models of the structure of composites. The ANSYS Workbench package was used as a calculation simulation platform. The true stress–strain curves for tension, Poisson’s ratios, and ultimate stresses for composite materials were obtained using the finite element method based on the micromechanical approach at the first stage. These values were calculated based on the stretching diagrams of the matrix and fillers and the condition of the ideality of their joint operation. At the second stage, the processes of mechanical pressing of composite materials were modelled based on their elastic–plastic characteristics from the first stage. The result is an assessment of the accumulation of residual strains at the stage before sintering. The degree of increase in total strain capability of composite materials after sintering was shown.

## 1. Introduction

Pressing and sintering are the main technological operations in powder metallurgy [1]. Traditional and modified types of these operations are sufficiently studied for various classes of powder and composite materials with the help of experimental studies, which are highlighted in many works [2,3,4,5,6,7,8,9]. For example, Wang Q.B. et al. [2] established the optimal parameters of the hot vacuum pressing process for obtaining a polymer composite material based on poly(ether ketone ketone) with high conductive and mechanical properties. The review [3] summarized the theoretical and practical research results on cold sintering processes used for various materials. In paper [4], joint sintering of ceramic and thermoplastic materials was experimentally studied using the concept of cold sintering. The authors of works [5,6,7,8] paid attention to the development and research of additive technologies that combine the advantages of traditional pressing and sintering processes. Zhang X. et al. [9] proposed a novel strategy for constructing a three-dimensional continuous graphene network architecture in a copper matrix based on powder metallurgy methods.

This work focuses on the composite materials with high physical and mechanical properties for work in conditions of intensive wear. The main criteria laid down in the development of such materials are the ability to work without lubrication, reduced wear of the part itself and the connected surface, resistance to the chemical influence of aggressive environments, reliable operation at low temperatures [10]. As a polymer matrix, which most fully satisfies the above criteria, polytetrafluoroethylene (PTFE) was chosen due to its unique properties [11]. It is an indispensable material in compressors that must ensure high purity of compressed gases and is used in the chemical, food, pharmaceutical, and other industries [12]. These materials have a wide range of achievable properties, which is ensured by using various technological techniques for obtaining the initial components [13] and the subsequent synthesis of the phases of the composition [14]. Due to the high viscosity of the melt, thermoplastic PTFE and compositions based on it cannot be processed into a product by worm extrusion or injection molding. Therefore, the methods of processing compositions based on PTFE into a composite are based on a two-stage process: obtaining a blank by pressing and subsequent heat treatment of the workpiece [15]. A special place in this plan belongs to the technology of pressing the composition into the workpiece, as the main operation of shaping the workpiece and structuring the composite [16,17]. In the process of further sintering of pressed materials at the fusion stage of matrix particles and fillers, the finished product’s chemical, physical, mechanical, and operational properties are established [18,19,20,21].

The importance of the results of experimental research [22,23,24] in recent years is combined with numerical modeling and simulation of pressing and sintering processes [25]. They help to reveal the influence of the main characteristics of materials and parameters of technological operations on the production process and the final operational properties of the sintered material.

In particular, Shaohua Chen et al. [26] studied the compaction mechanisms and factors affecting this process and developed a computational model for solid-state sintering with hot pressing. The novelty of their model lies in the combination of the first-order compaction model (the J–W model) and the kinetic Monte Carlo sintering method already developed.

Nosewicz S. et al. [27,28] presented original numerical models of the powder metallurgy process of a two-phase powder mixture NiAl–Al_2_O_3_ by the discrete element modeling. All technological stages were analyzed for obtaining composite materials, experimental results for the hot pressing were received, and microscopic studies of the NiAl–Al_2_O_3_ interface were conducted.

In the paper [29], a comprehensive approach was implemented to model the processes of obtaining sintered gears by powder metallurgy. The main goal of the work is a quantitative description of the reaction of the material during the production of the part at each stage. The discrete element method in the LIGGGHTS-PUBLIC software was used to simulate pressing, and the ABAQUS program and the Monte Carlo kinetic method were used for sintering. The Monte Carlo method was also used in works [30,31] for computational modeling of the sintering process.

In [32], the authors developed a simulation of the process of cold isostatic pressing of SiC powder using the Drucker–Prager–Cap model in the ABAQUS/Explicit1 software.

The sintering process of the two-component system “Al-graphene” was simulated by the method of molecular dynamics in [33] to study the behavior of graphene and its influence on the strength properties of the composite material.

In this work, the computer simulation was carried out using the ANSYS software package, which is available and widely used by scientists to simulate the process of pressing [34,35] and sintering [36,37]. However, insufficient attention has been paid to modeling the structure and properties of composite materials, that is, their behavior during pressing and sintering.

Publications on research results related to computer modeling of PTFE-composites by the technological pressing method are rare. This determines the scientific novelty of this work. An effective computer simulation of the pressing of PTFE-composites should allow us to accurately analyze their production process by pressing and the change in their behavior in a two-component composite material during the sintering process.

The work aims to estimate the general strain capacity of the sintered material with similar data after mechanical processing based on the simulation of micromechanical submodels to obtain real stretching diagrams of materials and the pressing process itself to assess the accumulation of residual strains to reduce the general strain capability. The ability to determine such residual strains before and after sintering makes it possible to assess the degree of influence of heat treatment on the general strain capability of materials, which is a valuable characteristic of the developed model.

To achieve this goal, first, solid geometric models of cubic volumes of composite materials randomly reinforced with spherical and short cylindrical inclusions were developed. Second, the three-dimensional elasto-plastic problems of stretching micromechanical models were solved using the finite element method, with obtaining valid stretching diagrams. Third, three-dimensional elasto-plastic problems of pressing composite materials with pre-obtained properties were solved using the finite element method based on the macromechanical model of pressing upper and lower punches in a matrix of hollow cylindrical samples from composite materials with obtaining residual strains, which means the general strain capability based on real stretching diagrams. Finally, an assessment of the increase in the mentioned general strain capability after heat treatment based on the experimental data was carried out.

## 2. Materials and Methods

### 2.1. Materials

The structure and properties of polymer composite materials were modelled in the work, the matrix of which was PTFE, with two types of fillers: spherical and short cylindrical inclusions. The initial data on the properties of these materials are presented in Table 1.

### 2.2. Technological Process of Obtaining Composite Materials

#### 2.2.1. Pressing

Pressing technology is the pressing of discrete materials that are compacted in a press under pressure to obtain a blank or product with a given size, shape, and density. This ensures the effectiveness of obtaining composites with the necessary operational properties.

The compacted material is a body consisting of discrete structure-forming elements (phases), mutually oriented in a certain way and packed in its volume with the formation of mechanical, adhesive, and other bonds in the places of mutual contact between PTFE particles and the filler [41].

The essence of pressing a discrete material is similar to the pressing of a solid body, with the difference that the ability of a solid body to deform in the transverse direction is replaced by the ability to move inward due to the reduction of porosity. During pressing, the volume of a discrete body changes from filling the cavities between the particles due to their displacement and plastic deformation.

Pressing was carried out on a hydraulic press MS-500 (laboratory of Sumy State University, Sumy, Ukraine).

The technological process of pressing includes the following stages [42]: press and press-form preparation; pouring the powder into the press-form; preliminary pressing; exposure at maximum pressing pressure; pressure relief; removing the tablet from the press-form (Figure 1).

At the first stage, the compaction of the powder of the PTFE-composition occurs due to the redistribution of the powder particles and their filling of the pores formed during the free filling of the material powder. Such compaction is not accompanied by plastic deformation of the powder particles of the composition. This stage is characterized by a significant effect of elastic unloading of some contacts, which occurs at the very beginning of the process of compacting the powder mass of the composition. Such local unloading of a part of the contacts reduces the forces of the adhesive bond between them. Moreover, it reduces the contact area of the corresponding contacts between the particles and, therefore, the occurrence of contact stresses [42].

The second stage of the process is characterized by the fact that the powder particles of the composition, packed as tightly as possible, provide a certain resistance to compression, the pressing pressure increases, and the density of the powder body does not increase for some time. At the same time, due to the elastic deformation of the particles, the role of local unloading of the contacts is insignificant, and the plastic deformation in the near-contact zone has a limited local character.

When the pressure exceeds the powder particles’ compression resistance, and their plastic deformation begins, the compaction process of the powder composition enters the third stage. From this moment, the plastic deformation covers the entire volume of each particle of the phases of the PTFE-composition, the displacement of the contacts stops, and they are fixed or destroyed when the values of the contact stresses are reached above the strength limit.

After pressing, elastic and highly elastic stresses appear in the compressed tablet, which is revealed: the first, when the pressing pressure is removed; the second, during storage and during sintering of the workpiece. The manifestation of elastic stresses is confirmed by the fact that after removing the pressing pressure, the tablet size increases by 2–3%.

#### 2.2.2. Sintering

The final stage of the production of PTFE-composites is the heat treatment of the workpieces — mainly free sintering in thermal furnaces, tablets removed from the press forms (Figure 2). Sintering was carried out on a thermal batch furnace (laboratory of Sumy State University, Sumy, Ukraine).

During sintering, the following processes occur sequentially in the compressed composition, which causes a change in its volume [42]:

Thermal expansion of the material, which takes place during almost the entire heating cycle up to 370 ± 10 °C;

The transition of the crystalline phase of PTFE into an amorphous phase, which is accompanied by an increase in the volume of the tablet by 25% at temperatures above 327 °C;

Thermal relaxation of highly elastic stresses obtained in the tableting process to temperature 327 °C;

Fusion of individual particles of material into a solid monolith, which is accompanied by the elimination of voids between particles and the latter sticking together, which leads to a certain decrease in the tablet’s volume. The emergence of highly elastic stress during tablet sintering is manifested in the fact that the blank, after cooling decreases in volume by 5–8%.

At the base of the process of polymer particles sticking together during heat treatment is a diffusion-segmentation mechanism that occurs over time, so its implementation requires a certain time interval. The particles begin to stick together when the polymer transitions from the crystalline phase to the amorphous phase. However, the segmentation mobility of macromolecules is not yet sufficient for segment diffusion to begin, so a slight increase in temperature is required 30–40 °C higher than the melting temperature of crystallites (327 °C). Before the temperature rises, the complete clarification of the polymer does not yet occur. Still, with an increase in the amount of heat supplied, the mobility of the segments in the macromolecules increases, and their active interaction with each other takes place, which is accompanied by the displacement of the existing cavities and a certain decrease in the volume of the tablet, which is externally manifested in achieving complete transparency of the tablet. Thus, the process of particles sticking together is carried out in a certain time interval, which requires exposure of the polymer at maximum temperatures, which ensures the most significant mobility of segments of macromolecules.

In the process of tablet sintering, the future product’s chemical, physical, mechanical, and electrical properties are laid down at the fusion stage of polymer particles. Therefore, sintering is the most critical stage of the technological process of manufacturing products. After keeping the tablet at the maximum temperature during its cooling stage, a supramolecular structure of the polymer is formed, which also significantly affects the properties of the future product.

## 3. Numerical Method

The first step is to determine the effect of heat treatment, which is preceded by mechanical treatment, based on the basic principles of building three-dimensional structural models of composites based on statistical data on the mechanical properties (elastic, plastic, and strength) of inclusions and matrix, finite element models of composites are built in the ANSYS Workbench program. A stress–strain diagram is obtained by simulating the stretching process of a sufficiently small element of a cubic-shaped composite [43,44,45]. The material’s strength and bearing capacity are considered with an analysis of the accumulation of damage in the model [46,47]. In structural models, the mechanical interaction of the microstructures of the filler and the matrix is simulated. This approach is called structural or micromechanical and has been actively developing recently [48,49,50,51].

The second step is to build simulation finite element models of a macroscopic hollow cylinder, loaded with pressure from one face and fixed to all others from linear movements. Here, the mechanical processing of the composite is simulated before thermal processing. Note that the materials of the hollow cylinders correspond to the stress–strain diagrams obtained in the first step. From the simulations, relative linear residual deformations were obtained, which identify the current accumulation of plastic deformations after mechanical processing; this means the average position of the point on the stress–strain diagram, the general residual strain capability, and ultimate strength. These obtained data are compared with the experimental data for the composite after heat treatment.

In the simulation, it is assumed that the contact surfaces of the matrix with fillers move together without separation or slippage, i.e., bonded contact. Moreover, the behavior of component materials is described by experimental stress–strain diagrams published in [38,39,40].

In the tensile deflected mode, in the simulation of the first one, the linear separation of the two opposite faces of the cube is ensured while maintaining their parallelism. When individual elements of the composite (first the filler and later the matrix) reached the ultimate strength limit in the simulation, it was considered that the element lost stiffness. This is modelled by the rapid growth of the plastic deformation component with a slight increase in load, so the limit of the strength of the composite is determined by the beginning of the cube’s loss of bearing capacity. This approach makes it possible to avoid considering the kinetics of material destruction because the local values of the parameters of the stress–strain state of the composite components often reach the limit values at the initial stages of loading the composite, but this does not lead to the exhaustion of its bearing capacity.

### 3.1. Construction of a Solid-State Model of a Composite Material

To numerically determine the elastic properties, the composite material randomly reinforced with spherical, long, and short cylindrical inclusions are simulated by the finite element method in a three-dimensional setting. The matrix of the composite material is industrial PTFE with the stress–strain diagram shown in Figure 3 [38] and Poisson’s ratio ν = 0.45. The filler with spherical inclusions is finely dispersed coke with the stress–strain diagram shown in Figure 4 [39], and Poisson’s ratio ν = 0.30. The filler with short cylindrical inclusions is kaolin (flakes) with a stress–strain diagram, as shown in Figure 5 [40], and Poisson’s ratio ν = 0.18. The geometric shape of the spherical filler is a sphere with diameters *d_i_* from 10 to 50 μm. Straight circular cylinders with diameters *d_i_* from 8 μm to 10 μm and length *l_i_* from 0.70 μm to 0.87 μm were taken for the short cylindrical filler. 

Remarks: It was believed that at deformations higher than the maximum ones in the stress–strain diagrams, the material loses its bearing capacity: the stiffness of the finite element becomes infinitely small.

A cube with an edge length of *a* = 200 μm for a spherical filler and *a* = 50 μm for a short cylindrical filler, with the properties of a matrix saturated with spherical or cylindrical inclusions with the properties of the filler is accepted as a solid-state model (Figure 6). The number of filler bodies was taken for reasons of achieving its volume content in the composite of 24.12% for spherical and 1.87% for short cylindrical.

#### 3.1.1. The Spherical Filler

The spherical filler is modelled by lottery with a uniform probability of the current diameter of the sphere *d_i_* in the range from 10 μm to 50 μm, respectively, and the position of the center *(x_i_; y_i_; z_i_)* in the range from *d_i_/2* μm to *a − d_i_/2* μm, which made it possible not to consider the issue of modeling the dissected filler. In addition, the condition of non-intersection of filler balls is imposed, which is mathematically formulated:(1)$$\sqrt{{\left(x_{i}-x_{j}\right)}^{2}+{\left(y_{i}-y_{j}\right)}^{2}+{\left(z_{i}-z_{j}\right)}^{2}}>\frac{d_{i}+d_{j}}{2},\;j=1\ldots \left(i-1\right)$$
where *i* – the number of the current ball, *j* – the number of the ball already accepted for construction.

#### 3.1.2. The Short Cylindrical Filler

The short cylindrical filler is modelled by lottery with a uniform probability of the current diameter of a straight circular cylinder *d_i_* in the range from 8 μm to 10 μm, respectively, the length *l_i_* from 0.70 μm to 0.87 μm, the position of the center *(x_i_; y_i_; z_i_)* in the range from *d_i_/2* μm to *a − d_i_/2* μm, and direction cosines *(a_xi_; a_yi_; a_zi_)*. Since *d_i_* >> *li*, two conditions are considered in the simulation:

Cylinders do not intersect with the bases;

Cylinders do not intersect with side surfaces.

The first condition is satisfied if the distances from the centers of the circle-bases of the cylinders with coordinates $\left(\begin{array}{ccc} x_{i}\pm a_{x_{i}}\cdot \frac{l_{i}}{2}; & y_{i}\pm a_{y_{i}}\cdot \frac{l_{i}}{2}; & z_{i}\pm a_{z_{i}}\cdot \frac{l_{i}}{2} \end{array}\right)$ and $\left(\begin{array}{ccc} x_{j}\pm a_{x_{j}}\cdot \frac{l_{j}}{2}; & y_{j}\pm a_{y_{j}}\cdot \frac{l_{j}}{2}; & z_{j}\pm a_{z_{j}}\cdot \frac{l_{j}}{2} \end{array}\right)$ to the straight lines of intersection of the planes of the circle-bases of the cylinders described by the equations
(2)$$\frac{x-\frac{\left|\begin{array}{cc} a_{xi}\cdot x_{i}+a_{yi}\cdot y_{i}+a_{zi}\cdot z_{i} & a_{yi}\\ a_{xj}\cdot x_{j}+a_{yj}\cdot y_{j}+a_{zj}\cdot z_{j} & a_{yj} \end{array}\right|}{\left|\begin{array}{cc} a_{xi} & a_{yi}\\ a_{xj} & a_{yj} \end{array}\right|}}{\left|\begin{array}{cc} a_{yi} & a_{zi}\\ a_{yj} & a_{zj} \end{array}\right|}=\frac{y-\frac{\left|\begin{array}{cc} a_{xi} & a_{xi}\cdot x_{i}+a_{yi}\cdot y_{i}+a_{zi}\cdot z_{i}\\ a_{xj} & a_{xj}\cdot x_{j}+a_{yj}\cdot y_{j}+a_{zj}\cdot z_{j} \end{array}\right|}{\left|\begin{array}{cc} a_{xi} & a_{yi}\\ a_{xj} & a_{yj} \end{array}\right|}}{\left|\begin{array}{cc} a_{zi} & a_{xi}\\ a_{zj} & a_{xj} \end{array}\right|}=\frac{z}{\left|\begin{array}{cc} a_{xi} & a_{yi}\\ a_{xj} & a_{yj} \end{array}\right|}$$
do not simultaneously exceed their respective radii:(3)$$\begin{array}{l} \begin{array}{cccccc} a_{x1}=\left|\begin{array}{cc} a_{yi} & a_{zi}\\ a_{yj} & a_{zj} \end{array}\right|; & a_{y1}=\left|\begin{array}{cc} a_{zi} & a_{xi}\\ a_{zj} & a_{xj} \end{array}\right|; & a_{z1}=\left|\begin{array}{cc} a_{xi} & a_{yi}\\ a_{xj} & a_{yj} \end{array}\right|; & & & \end{array}\\ \begin{array}{ccc} x_{1}=\frac{\left|\begin{array}{cc} a_{xi}\cdot x_{i}+a_{yi}\cdot y_{i}+a_{zi}\cdot z_{i} & a_{yi}\\ a_{xj}\cdot x_{j}+a_{yj}\cdot y_{j}+a_{zj}\cdot z_{j} & a_{yj} \end{array}\right|}{\left|\begin{array}{cc} a_{xi} & a_{yi}\\ a_{xj} & a_{yj} \end{array}\right|}; & y_{1}=\frac{\left|\begin{array}{cc} a_{xi} & a_{xi}\cdot x_{i}+a_{yi}\cdot y_{i}+a_{zi}\cdot z_{i}\\ a_{xj} & a_{xj}\cdot x_{j}+a_{yj}\cdot y_{j}+a_{zj}\cdot z_{j} \end{array}\right|}{\left|\begin{array}{cc} a_{xi} & a_{yi}\\ a_{xj} & a_{yj} \end{array}\right|}; & z_{1}=0; \end{array}\\ \delta_{i}=\sqrt{\frac{{\left|\begin{array}{cc} a_{y1} & a_{z1}\\ y_{1}-y_{i} & z_{1}-z_{i} \end{array}\right|}^{2}+{\left|\begin{array}{cc} a_{z1} & a_{x1}\\ z_{1}-z_{i} & x_{1}-x_{i} \end{array}\right|}^{2}+{\left|\begin{array}{cc} a_{x1} & a_{y1}\\ x_{1}-x_{i} & y_{1}-y_{i} \end{array}\right|}^{2}}{a_{x1}{}^{2}+a_{y1}{}^{2}+a_{z1}{}^{2}}};\\ \delta_{j}=\sqrt{\frac{{\left|\begin{array}{cc} a_{y1} & a_{z1}\\ y_{1}-y_{j} & z_{1}-z_{j} \end{array}\right|}^{2}+{\left|\begin{array}{cc} a_{z1} & a_{x1}\\ z_{1}-z_{j} & x_{1}-x_{j} \end{array}\right|}^{2}+{\left|\begin{array}{cc} a_{x1} & a_{y1}\\ x_{1}-x_{j} & y_{1}-y_{j} \end{array}\right|}^{2}}{a_{x1}{}^{2}+a_{y1}{}^{2}+a_{z1}{}^{2}}};\\ \bar{\begin{array}{ccc} \delta_{i}\leq \frac{d_{i}}{2} & \wedge & \delta_{j}\leq \frac{d_{j}}{2} \end{array}} \end{array}$$
where *i* is the number of the current cylinder, *j* is the number of the cylinder already accepted for construction.

The second condition is sufficiently satisfied if, at the same time, the angle *α* between the generating cylinders does not exceed the minimum angle formed if the cylinders touch at one point, and at the same time, the centers of masses of the cylinders are too close to each other:(4)$$\begin{array}{l} \alpha =\arccos\left(a_{x_{i}}\cdot a_{x_{j}}+a_{y_{i}}\cdot a_{y_{j}}+a_{z_{i}}\cdot a_{z_{j}}\right);\\ \alpha_{\min}=\arccos\left(\frac{d_{i}\cdot d_{j}-l_{i}\cdot l_{j}}{\sqrt{d_{i}{}^{2}+l_{i}{}^{2}}\cdot \sqrt{d_{j}{}^{2}+l_{j}{}^{2}}}\right);\\ \delta_{\min}=\sqrt{{\left(\frac{d_{i}}{2}\right)}^{2}+{\left(\frac{l_{i}}{2}\right)}^{2}+{\left(\frac{d_{j}}{2}\right)}^{2}+{\left(\frac{l_{j}}{2}\right)}^{2}+2\cdot \cos\alpha \cdot \sqrt{\left({\left(\frac{d_{i}}{2}\right)}^{2}+{\left(\frac{l_{i}}{2}\right)}^{2}\right)\cdot \left({\left(\frac{d_{j}}{2}\right)}^{2}+{\left(\frac{l_{j}}{2}\right)}^{2}\right)}};\\ \delta_{ij}=\sqrt{{\left(x_{i}-x_{j}\right)}^{2}+{\left(y_{i}-y_{j}\right)}^{2}+{\left(z_{i}-z_{j}\right)}^{2}};\\ \bar{\begin{array}{ccc} \alpha \leq \alpha_{\min} & \wedge & \delta_{ij}\leq \delta_{\min} \end{array}}. \end{array}$$
where *i* is the number of the current cylinder, *j* is the number of the cylinder already accepted for construction, *δ_ij_* is the distance between the centers of mass of the cylinders.

It should be noted that the second condition is excessively strict and excludes potentially suitable cases of collocation of cylinders. The set of requirements turns out to be suitable for use due to the small volume content of the filler.

### 3.2. Construction of a Finite Element Model of a Composite Material

#### 3.2.1. Micromechanical Approach

The solid model is divided into finite three-dimensional elements at the first stage of the micromechanical approach. The solid material model computational mesh is created with the ANSYS Meshing program using the tetrahedron method having applied the Patch Conforming algorithm. This method is preferred for relatively complex geometries such as grooves, channels, and corners with angles. Tetrahedrons can generate more cells than quadratic elements of the equivalent mesh size. Whereas the Patch Conforming algorithm is due to use, once the geometry of small details is under concern. Meshes of both micromechanical and macromechanical models are assessed by three main quality criteria Mesh Orthogonality, Aspect Ratio, and Expansion Factor, which ensure modelling accuracy and convergence.

Boundary conditions are applied as zero linear displacement on the area of one cube face in the normal direction and gradual ramped finite linear displacement on the area of the opposite cube face in the exterior normal direction (Figure 7). Weak Springs technology is used for otherwise not enough supported material models becoming unstable and moving. Once such an unstable structure is detected, “weak springs” are automatically created to make it capable of withstanding small external forces. A linear (uniaxial) deflected mode – tension has been modelled. All volumes of the modelled matrix and fillers touch with ideal contact, i.e., nodes of elements on joint surfaces have the same displacement (Figure 8).

#### 3.2.2. Macromechanical Approach

At the second stage, a macromechanical approach is applied to the simulation of mechanical pressing of workpieces of composite materials in a press form according to Figure 9. The hollow cylinder mesh is built using the sweep method, capable of the model’s detailed mesh construction about the rotation axis, for which the input and output boundaries have the same topology [52]. The finite element model of the hollow cylinder is rigidly clamped behind the lower base, and the side faces are constrained in radial linear movements. For the second stage, the elastic properties of composite materials are obtained in the form of stress–strain curves and Poisson’s ratio at the first stage. The load is applied with a pressure of 35 MPa on the upper face.

## 4. Results and Discussion

### 4.1. Model Simulation Results

According to von Mises stress for matrices and fillers, equivalent stresses at failure for the first stage are shown in Figure 10 and Figure 11. And stress–strain curves for simulations of micromechanical models of composites are shown in Figure 12 and Figure 13.

Equivalent stresses and residual (plastic) relative linear strains at full compression for the second stage according to von Mises stress for composites are shown in Figure 14 and Figure 15.

### 4.2. Verification of the Results of the Micromechanical Approach (First Stage)

Mesh independence studies are performed to determine the results’ dependence on the mesh density of both stages. The average element size has varied over three iterations for each model. The percent relative errors for maximal equivalent Von-Mises stresses are less than 10% between the first and second size reductions and less than 1.5% between the second and third, which is satisfactory since a significant further increase in nodes obviously does not lead to a significant increase in a stress state.

In general, it turns out that simulation using the finite element method at the first stage predicts the strength of the considered materials well (Table 2). The strength limit is determined at the inflection point of the stress–total strain curves (Figure 12 and Figure 13a), where the loss of the load-bearing capacity of the volume of the cubic form made of composite materials of various fillers is postulated, with the subsequent dramatic development of the destruction processes. Considering the actual diagrams make it possible to significantly refine the prediction of strength limits and Poisson’s ratios and obtain calculated stress–total strain curves for the simulation of the mechanical processing of materials.

### 4.3. Verification of the Results of the Macromechanical Approach (Second Stage)

During the second stage, the volumes of the hollow cylinders are compressed. Finite element simulations make it possible to determine residual plastic strains at the end of pressing in the press form shown in Figure 9a under sufficient time to develop plastic strains because a static elastic-plastic three-dimensional problem is being solved. These residual plastic strains are subtracted from the general strain capacity of the composites, which makes it possible to compare them with such capacity after heat treatment (Table 3).

The general strain capacity after mechanical processing consists of the total strain capacity of the material before this mechanical processing minus the accumulated plastic deformation during mechanical processing.

## 5. Conclusions

In the work, a computer simulation of the mechanical behavior of polymer composite materials under pressing the workpiece has been implemented to determine the general strain capacity of the unsintered material, followed by a comparison of this indicator after the sintering of the material. The ability to determine such residual strains before and after sintering has made it possible to assess the degree of influence of heat treatment on the general strain capacity of materials, which is a valuable characteristic of the developed model.

In the first stage, to determine the effect of heat treatment preceded by mechanical treatment, based on statistical data on the mechanical properties of inclusions and the matrix, finite element models of composites have been built in the ANSYS Workbench program. Structural models based on a polymer matrix (PTFE) and two types of fillers with different shapes have been simulated: spherical inclusions (coke) and short cylindrical inclusions (kaolin) with mechanical interaction between the microstructures of the filler and the matrix. A stress–strain diagram has been obtained by further simulating the stretching process of a sufficiently small element of a cubic-shaped composite, and the material’s strength and bearing capacity have been evaluated with an analysis of the accumulation of damage in the model.

In the second stage, the mechanical processing of the composite is simulated before thermal processing. For this purpose, simulation finite element models of a macroscopic cylinder with a hole, loaded with pressure from one face and fixed to all others from linear movements, have been built. From the simulations, relative linear residual strains have been obtained, identifying the current accumulation of plastic deformations after mechanical processing: average point position on the stress–strain diagram, the general residual strain capability, and ultimate strength. These obtained data have been compared with the experimental data for the composite after heat treatment.

It is found that the finite element simulation at the first stage predicts the strength of the considered materials well. In particular, the relative uncertainty beyond the strength limits of the simulations compared to the experiment for PTFE + coke is 6.02%, and for PTFE + kaolin is 0.33%.

The results of the macromechanical approach (second stage) have made it possible to obtain the average level of accumulation of equivalent plastic deformations and to estimate the increase in the general strain capacity after heat treatment of materials during stretching: PTFE + coke after heat treatment increases the ability to deform by 1.3 times, and PTFE + kaolin - by 15.4 times, which is obviously related to the polymerization of the matrix additionally strengthened by low saturation with the filler.

Weak Springs simulation technology has brought the simulation models closer to the real tension process. The micromechanical model has been made on the basis of the assumptions of an ideal cohesion between the matrix and the filler, which can be refined by considering a non-ideal contact based on cohesive zone modelling simulation technology. The considered macromechanical model has not considered the friction phenomenon between the press mandrel and the composite, which can lead to the inability to simulate the destruction of the rubbing surface.

Further research is related to applying the obtained algorithm for building a computer simulation on composite materials with a different type of filler (a form of inclusions), for example, long cylindrical inclusions, etc. It is also interesting to study the composite material’s behavior during the sintering process using computer simulation in the ANSYS software.

The principles of building the obtained models using a micro- and (or) macro-mechanical approach will allow scientists in the field of polymer composite materials technology to apply them to predict the patterns and dependencies of the behavior of two-component materials and, therefore, their properties, in the conditions of technological operations of obtaining or designing new materials.

## Figures and Tables

**Figure 1 polymers-14-05288-f001:**
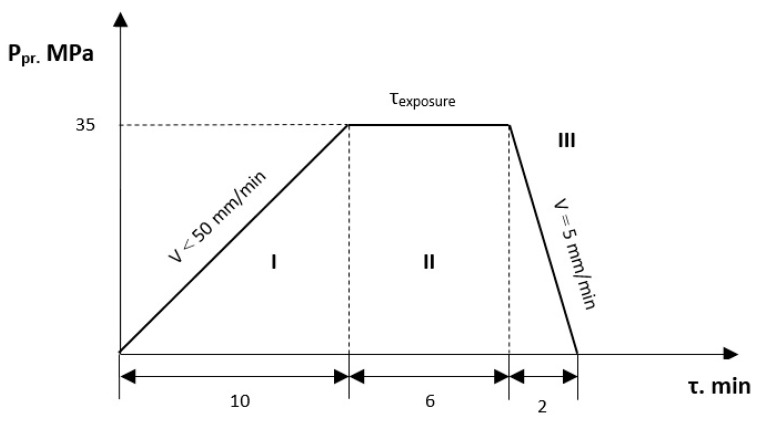
Graph of the real process of pressing PTFE-composition by the MS-500 press: I — pressure rise, II — endurance at maximum pressure, III — pressure relief.

**Figure 2 polymers-14-05288-f002:**
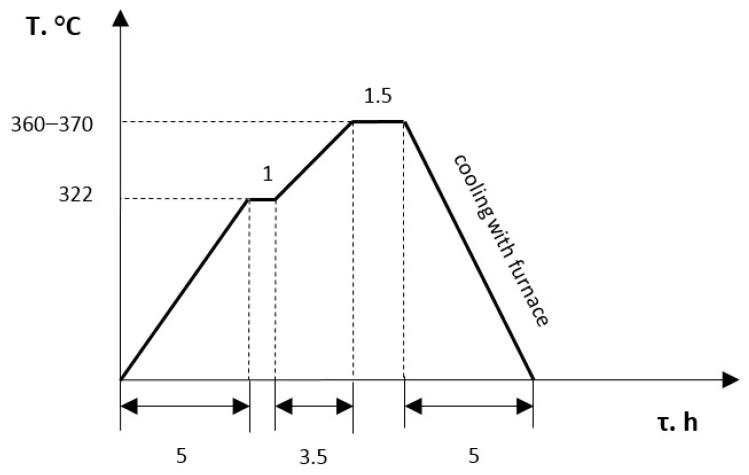
Graph of the real sintering process of PTFE-composition in induction furnace.

**Figure 3 polymers-14-05288-f003:**
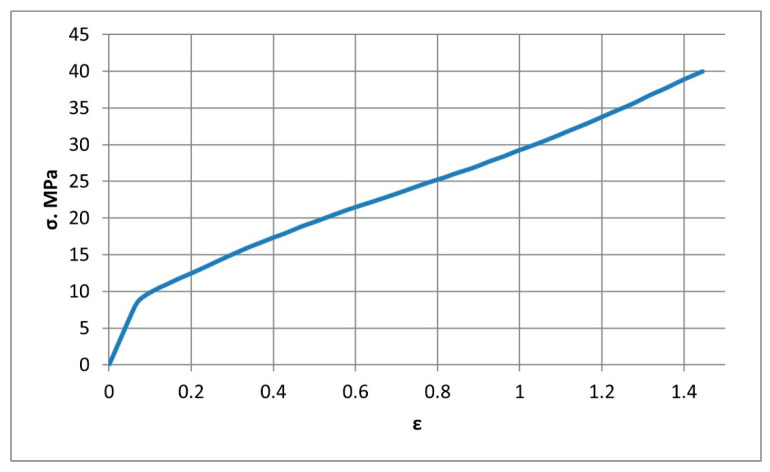
The true stress–strain curve for industrial PTFE at 25 °C, provided that the speed of linear elongation deformations does not exceed 500 μm/s.

**Figure 4 polymers-14-05288-f004:**
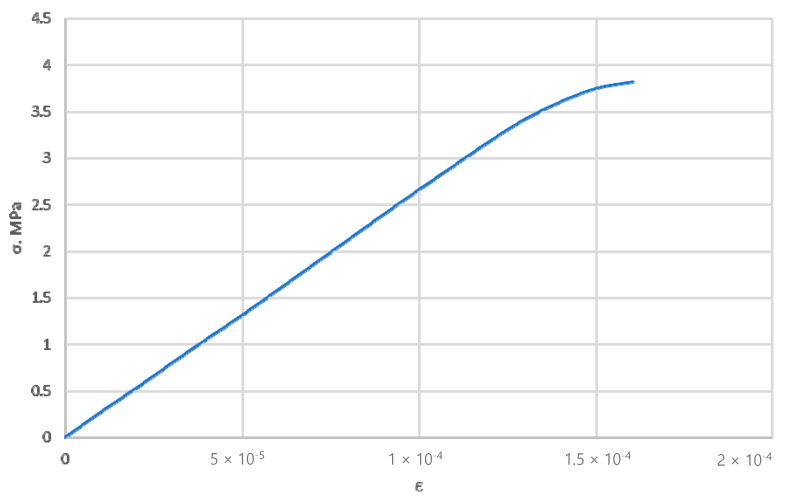
The true stress–strain curve for finely dispersed coke with zero porosity.

**Figure 5 polymers-14-05288-f005:**
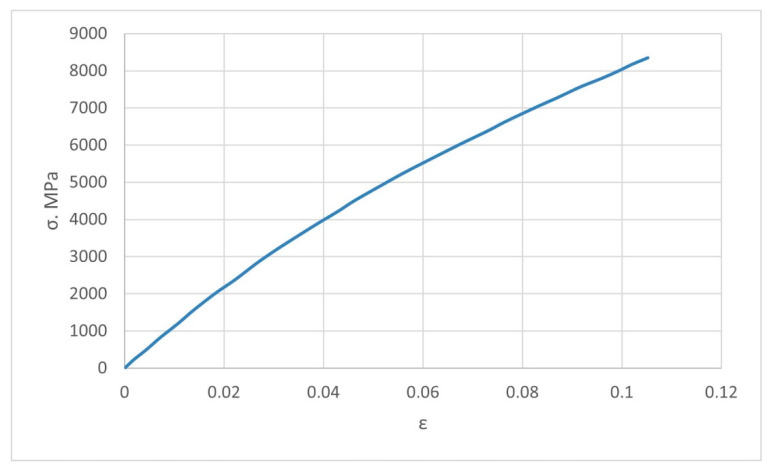
The true stress–strain curve for flaky kaolin.

**Figure 6 polymers-14-05288-f006:**
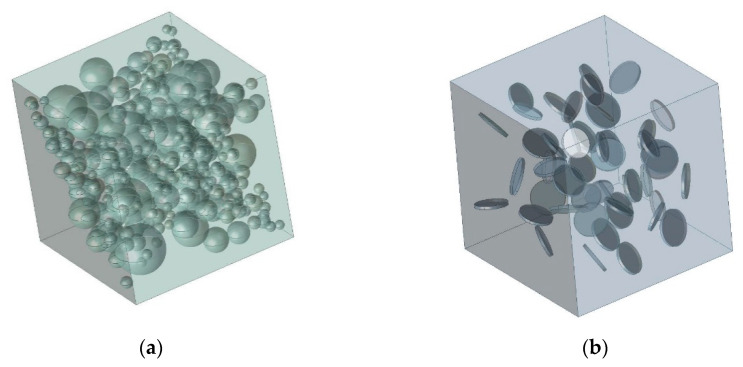
A solid-state model of composite materials randomly reinforced with spherical (**a**) and short cylindrical (**b**) inclusions.

**Figure 7 polymers-14-05288-f007:**
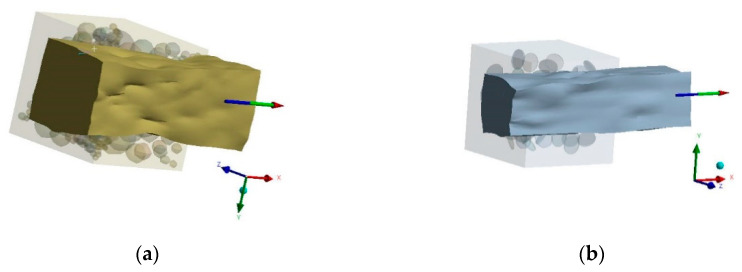
The deformation state of the finite element micromechanical model of composite materials randomly reinforced with spherical (**a**) and short cylindrical inclusions (**b**) with applied boundary conditions with undeformed model.

**Figure 8 polymers-14-05288-f008:**
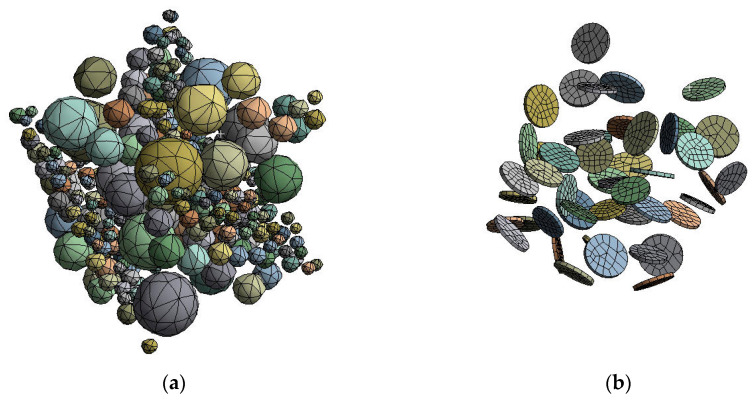
Finite element micromechanical models of fillers with spherical (**a**) and short cylindrical (**b**) inclusions.

**Figure 9 polymers-14-05288-f009:**
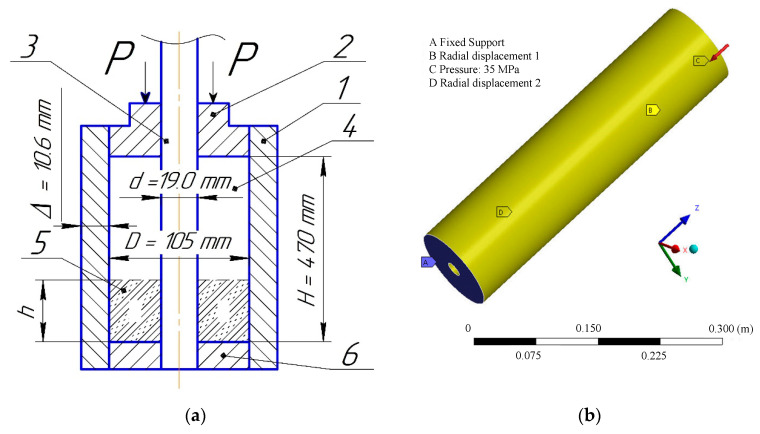
Macromechanical scheme of composite materials during compression pressing: 1 — matrix; 2, 6 — upper and lower punches; 3 — mandrel; 4 — powder weight; 5 — compressed tablet: experimental setup (**a**); solid-state simulation model (**b**).

**Figure 10 polymers-14-05288-f010:**
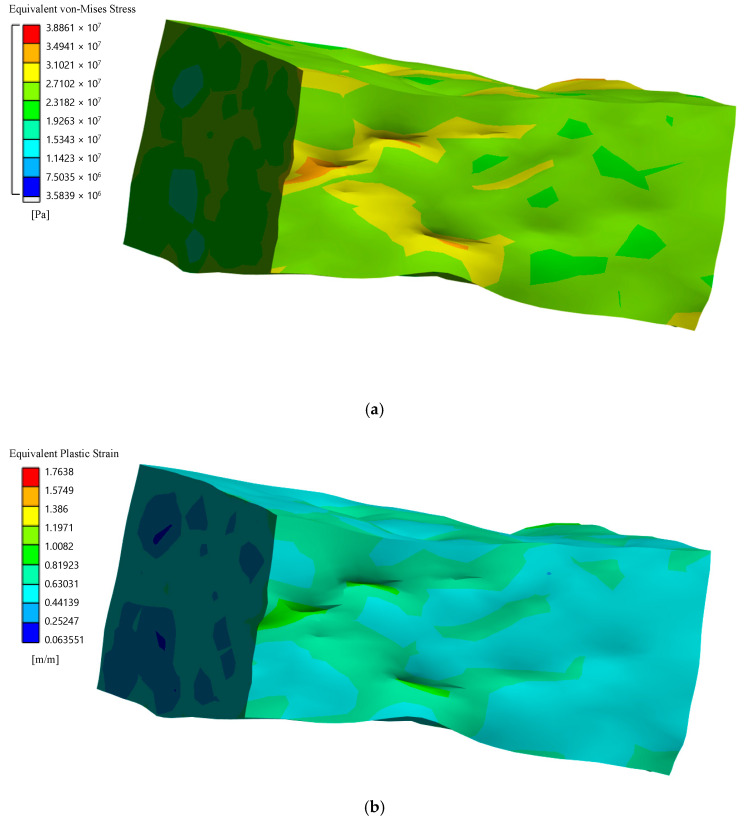
Equivalent von Mises stress, Pa (**a**) and equivalent plastic strain (**b**) with spherical filler at the moment of loss of bearing capacity (first stage).

**Figure 11 polymers-14-05288-f011:**
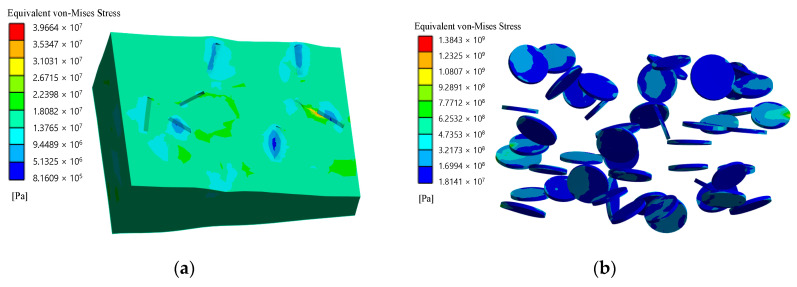
Equivalent stresses according to von Mises stress, Pa for matrix (**a**) and filler (**b**) and equivalent plastic strain (**c**) with short cylindrical filler at the moment of loss of bearing capacity (first stage).

**Figure 12 polymers-14-05288-f012:**
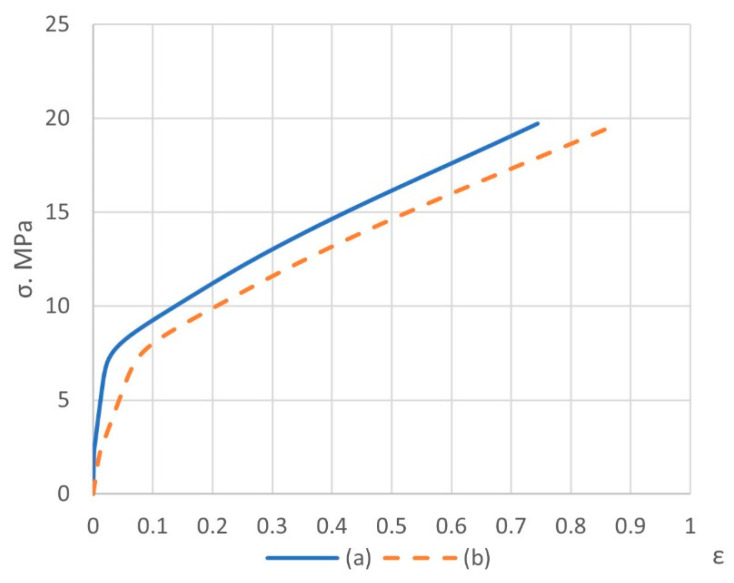
Stress– plastic strain (**a**) and stress– total strain (**b**) curves for composite with spherical filler (first stage).

**Figure 13 polymers-14-05288-f013:**
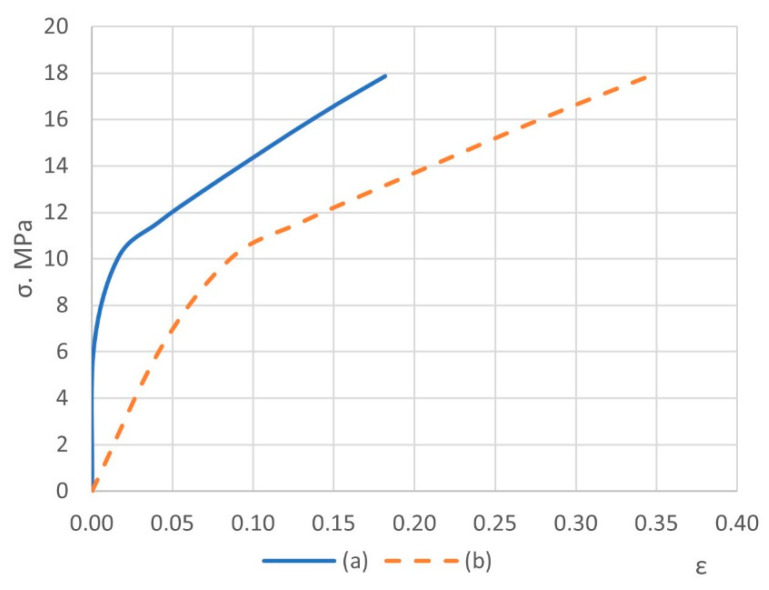
Stress– plastic strain (**a**) and stress– total strain (**b**) curves for composite with short cylindrical filler (first stage).

**Figure 14 polymers-14-05288-f014:**
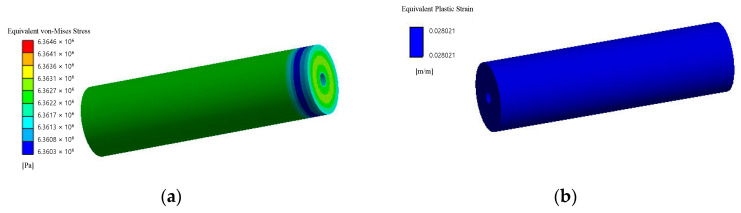
Equivalent stresses according to von Mises stress (**a**) and residual (plastic) relative linear strains (**b**) for composite with spherical filler at the moment of full compression (second stage).

**Figure 15 polymers-14-05288-f015:**
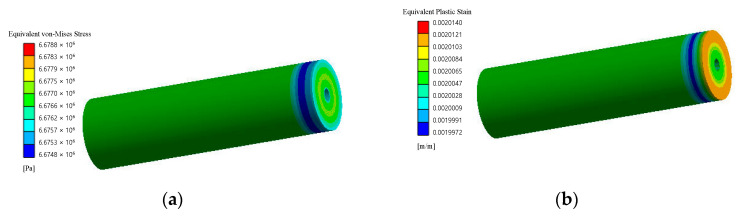
Equivalent stresses according to von Mises stress (**a**) and residual (plastic) relative linear strains (**b**) for composite with short cylindrical filler at the moment of full compression (second stage).

**Table 1 polymers-14-05288-t001:** Physical and mechanical properties of materials for research [38,39,40].

Designation	Material	Size, μm	Density, kg·m^−3^	Modulus of Elasticity, MPa	Poisson’s Ratio
Matrix	PTFE	50–500	2200	686.5	0.45
Spherical inclusions	Coke	10–50	1730	500	0.30
Short cylindrical inclusions	Kaolin	up to 10	2350	92000	0.18

**Table 2 polymers-14-05288-t002:** Characteristics of strength and elastic-plasticity for simulation models of composite materials with a micromechanical approach (first stage).

Composition (Matrix/Filler)	Mass/Volume Fractions of Matrix and Filler	The Shape of the Filler	Ultimate Strength Simulation/Experiment, MPa	Average Ultimate Total Strain, %	Average Ultimate Plastic Strain, %	Poisson’s Ratio
PTFE + coke	80:20/75.88:24.12	spherical	19.72/18.6	88.34	73.55	0.406 ± 0.005
PTFE + kaolin	98:2/98.13:1.87	short cylindrical	17.86/17.8	28.49	18.18	0.445 ± 0.005

**Table 3 polymers-14-05288-t003:** Strain capacity of composite materials after mechanical processing based on simulations and experimental data for materials after heat treatment.

Composition (Matrix/Filler)	After Mechanical Processing (Simulation Data)			After Heat Treatment (Experimental Data)
	Average Total Strain under 35 MPa Pressure, %	Average Plastic Strain under 35 MPa Pressure, %	Total Strain Capacity after Unloading, %	Total Strain Capacity, %
PTFE + coke	5.9844	2.8021	88.34 − 2.80 = 85.54	115
PTFE + kaolin	1.4363	0.20061	28.49 − 0.20 = 28.29	432

## Data Availability

The data presented in this study are available on request from the corresponding author.

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
