# Peer review of "Computer Simulation of Composite Materials Behavior under Pressing"

_polymers, 2022, doi:10.3390/polym14235288_

Round 1

Reviewer 1 Report

The following suggestions are important for the enhancement of the quality of this article furthermore. Thus, please do the needful. 

1. Expand PEKK in line 44. 

2. The selection of PTFE for the composite preparation has been explained in a less manner. therefore, Please add more additional justification for this section and also add furthermore relevant references. Especially, epoxy resin is the best matrix for the development of polymer composite but in your PTFE was used. Henceforth, justification is mandatory. 

3. Please add references for Table 1. Some of the data mentioned in Table 1 are different from the available data so references are mandatory. In particular, the density of Kaolin is obtained as 2650 kg/m^3 but you have mentioned as 2350 kg/m^3; also the density of Coke is obtained as 1000 kg/m^3 but you have mentioned as 1730 kg/m^3; Please clarify these differences in data. 

4. Since the major contents are deals with the primary processes in section 2.2, please add adequate references without fail. 

5. Spherical Filler and Short Cylindrical Filler are numerically modelled. But please add references or add additional content in this section in order to enhance the quality of this section. 

6. There is no clear explanation for the development of the finite element model and imposed boundary conditions in the micromechanical approach. Please provide the needful imposed mesh details and boundary conditions details in a clear manner. 

7. How did you obtain the value of input pressure 35 MPa in the micromechanical approach? Please justify your answer. 

8. Since you are work uses computational simulation as the primary methodology, have the authors conducted any reliability tests such as a grid independence study or experimental correlation study for this FEA computation? if yes please explain clearly the imposed verification contents along with needful references. 

9. Please enhance the qualities of computational structural outcomes and their relevant graphs (Figures 10 to 15)

10. Put the main title number as 6 in line 391.

Author Response

Dear reviewer! The authors are sincerely grateful to you for reading our paper and reviewing it.

The following suggestions are important for the enhancement of the quality of this article furthermore. Thus, please do the needful. 

  1. Expand PEKK in line 44.

It has been fixed.y

  1. The selection of PTFE for the composite preparation has been explained in a less manner. therefore, Please add more additional justification for this section and also add furthermore relevant references. Especially, epoxy resin is the best matrix for the development of polymer composite but in your PTFE was used. Henceforth, justification is mandatory.

The explanation about choice of polymer matrix PTFE was added.

  1. Please add references for Table 1. Some of the data mentioned in Table 1 are different from the available data so references are mandatory. In particular, the density of Kaolin is obtained as 2650 kg/m^3 but you have mentioned as 2350 kg/m^3; also the density of Coke is obtained as 1000 kg/m^3 but you have mentioned as 1730 kg/m^3; Please clarify these differences in data. 

The initial properties of objects for research are regulated by the relevant standards. They also vary depending on the specific brand of material. We added the referances for Table 1.

  1. Since the major contents are deals with the primary processes in section 2.2, please add adequate references without fail. 

All necessary reference was added.

  1. Spherical Filler and Short Cylindrical Filler are numerically modelled. But please add references or add additional content in this section in order to enhance the quality of this section. 

It has been fixed.

  1. There is no clear explanation for the development of the finite element model and imposed boundary conditions in the micromechanical approach. Please provide the needful imposed mesh details and boundary conditions details in a clear manner. 

It has been fixed.

  1. How did you obtain the value of input pressure 35 MPa in the micromechanical approach? Please justify your answer. 

An experiment was conducted in the laboratory at a pressure of 35 MPa. This value is standard when pressing PTFE-based composites.

  1. Since you are work uses computational simulation as the primary methodology, have the authors conducted any reliability tests such as a grid independence study or experimental correlation study for this FEA computation? if yes please explain clearly the imposed verification contents along with needful references. 

The explanation was added.

  1. Please enhance the qualities of computational structural outcomes and their relevant graphs (Figures 10 to 15)

The quality of Figs was improven.

  1. Put the main title number as 6 in line 391.

It was done.

Reviewer 2 Report

please read the attachment. Thank you.

Author Response

Dear reviewer! The authors are sincerely grateful to you for reading our paper and reviewing it.

All suggestions of Review 2 were corrected and added into paper text:

- Figure 1: the quality of this Figure is poor. Please increase its resolution. It has been improved.

- Section 3: "Numerical Approach Used for Simulation" should be considered changing to "Numerical Method." It has been done.

- Please combine Figures 3-5 into one Figure. Please revise the value of the vertical axis (Figure 5). They seem bigger than the others (in other figures). Unable to merge pic. 3, 4 and 5 into one, since the scales of the vertical axes are too different.

- Lines 298-299: please consider changing items 1,2 with items i, ii. It has been done.

- Figures 12-13: please combine these figures with being easy for comparison. It has been done.

- Section 4 and Section 5 should be combined into one section (Results and Discussion). It has been done.

- Figure 9: Add to the dots through Mandrel (3) to represent the center of the center, the symmetrical axis at the cross-section is defined (Figure 9. a) It has been done.

Please follow the journal template to choose Fig. X or Figure. X to mention or explain in the text. In this manuscript, in Figures 1-9, the authors noted "Fig. x," but in Figures 10-15, the authors used Figure. X to mention in the text. Please revise. It has been done.

- References: the is a shortage of concerns. Please consider adding the following novel articles to your literature review/ related works if suitable for your future experimental studies. New references were added.

- Have the authors experimented with verifying the numerical result? We have added an explanation in the article.

- Could the authors please introduce the workstations or desktop functions used to simulate your work? The simulation was carried out in the working program Ansys Workbench. The stages of modeling and their verification are presented in detail in the article.

- How did the authors validate the results? And add the main limitation of this study. We have added an explanation in the article.

Round 2

Reviewer 1 Report

Thanks for your support and action